# The structure balance of gene-gene networks beyond pairwise interactions

**Nastaran Allahyari**[1], **Amir Kargaran**[1], **Ali Hosseiny**[1], **G. R. Jafari**[1,2]*

**1** Department of Physics, Shahid Beheshti University, Tehran, Iran, **2** Institute of Information Technology and Data Science, Irkutsk National Research Technical University, Irkutsk, Russia

* g_jafari@sbu.ac.ir

## Abstract

Despite its high and direct impact on nearly all biological processes, the underlying structure of gene-gene interaction networks is investigated so far according to pair connections. To address this, we explore the gene interaction networks of the yeast *Saccharomyces cerevisiae* beyond pairwise interaction using the structural balance theory (*SBT*). Specifically, we ask whether essential and nonessential gene interaction networks are structurally balanced. We study triadic interactions in the weighted signed undirected gene networks and observe that balanced and unbalanced triads are over and underrepresented in both networks, thus beautifully in line with the strong notion of balance. Moreover, we note that the energy distribution of triads is significantly different in both essential and nonessential networks compared to the shuffled networks. Yet, this difference is greater in the essential network regarding the frequency as well as the energy of triads. Additionally, results demonstrate that triads in the essential gene network are more interconnected through sharing common links, while in the nonessential network they tend to be isolated. Last but not least, we investigate the contribution of all-length signed walks and its impact on the degree of balance. Our findings reveal that interestingly when considering longer cycles, not only, both essential and nonessential gene networks are more balanced compared to their corresponding shuffled networks, but also, the nonessential gene network is more balanced compared to the essential network.

## Introduction

Today, various studies investigate genomic information based on pairwise connections in gene interaction networks [1]. However, the interesting collective behaviors that emerge from these interactions can not be described by simply considering pairs of genes. In other words, while studying pair connections has well broadened our view on the functionality of genes, the higher-order organizations are yet to be explored. To be specific, studies demonstrate that genes are categorized into two main groups [2]. Functionally, essential genes play a more vital role in the biological process, and locally they form a denser network compared to nonessential genes. Yet the crucial question raised here is if there exists a structure beyond these pairwise interactions in these two networks. If so, what is the difference in the underlying structure

**Data Availability Statement:** Data is available from https://figshare.com/articles/dataset/Data_File_S3_Genetic_interaction_profile_similarity_matrices_zip/19350527.

**Funding:** The author(s) received no specific funding for this work.

**Competing interests:** The authors have declared that no competing interests exist.

between essential and nonessential networks? Suppose in a signed interaction network genes *A*, *B*, and *C* are connected, is it logical to consider the interaction *AB* detached from its context, that is, triad *ABC*? What is the impact of interactions *AC* and *BC* on the interaction between genes *A* and *B*? It is known that triadic interactions play a significant role in the construction of real-world networks [3, 4], and structural balance theory (*SBT*) has well discussed these interactions. In this work, we apply *SBT* to the gene interaction networks to answer the following questions: Is there a structure beyond pairwise interaction in the gene interaction networks? Which types of triads, balanced or unbalanced, are over (under) represented in these networks compared to the shuffled networks regarding both the frequency and the energy distributions? Is there a difference between essential and nonessential networks in the pattern of connection between triads? In addition, when considering all lengths of cycles, which network is more balanced? And do all genes have an equal impact on the final networks' degree of balance? These questions are the basis of this study.

*SBT* was introduced in social psychology by Heider to investigate the structure of tension in networks whose mutual relationships are explained in terms of friendship and hostility [5]. Later this theory has been generalized for graphs by Cartwright and Harary through considering the triads as low-dimensional motifs [6]. One of the standard applications provided by balance theory is to measure the degree of balance/ stability in networks [7–12]. On the other hand, quantifying the degree of unbalancing/ frustration in a signed network was proposed as well [13]. Similarly, in biological networks distance to the exact balance is computed [14–17]. Moreover, several researchers have studied the dynamics based on which an unbalanced network achieves balance through reducing unbalanced triads [18–25]. Some studies provide further theoretical expansion of balance theory employing methods from Boltzmann-Gibbs statistical physics to unravel the dynamics behind the structural balance [4, 26, 27]. An appealing application of balance theory recently applied predicts which correlation matrix coefficients are likely to change their signs in the high-dimensional regime [28]. Consequently, there have been two main trends in the literature of *SBT*: (1) Studying the analytical aspects theoretically [19, 29–35], (2) Applying it to a wide variety of real-signed social, economic, ecologic, and political networks empirically to clarify their structures [36–43]. Among these applications, it should be mentioned that understanding the structure entirely, not partially, calls for considering not only short-range interactions but also longer-range cycles [44–47]. Accordingly, we analyze the structural balance of gene interaction networks. We study the genetic interaction profile similarity matrices of the yeast Saccharomyces cerevisiae [48, 49], which has been categorized into two main classes, namely, essential and nonessential. Among all 5500 genes, approximately 1000 genes are essential because of their vital functional role in biological processes. According to the threshold taken by Costanzo and et al. in [48], essential genes have higher degrees and are considered hubs in the global network. Thus, these genes play a considerable role in the local structure of the network. On top of that, essential genes have higher prediction power compared to nonessential genes [50, 51].

Here, we investigate the weighted, signed, and undirected networks of genetic interaction for essential and nonessential genes of the yeast Saccharomyces cerevisiae. Primarily, we are interested in probing the existence of structure beyond the pairwise gene interactions in these networks. To this aim as in our previous study [52], we compare the spectrum of eigenvalues between genetic interaction matrices and their shuffled versions. The rest of the paper is organized as follows. First, we explore the frequency of triads in the gene networks according to the notion of over and under-representation of different types of triad compared to the shuffled networks. Afterward, we assign energy levels to unique configurations of triads and demonstrate triads' energy distributions. Then, the energy-energy mixing patterns between triads are analyzed to systematically investigate how triads with different energies are connected in the

networks. Additionally, we examine the balance of the gene interaction networks by considering all lengths of cycles. Last but not least, we propose a list of genes which have the highest degree of balance.

## Materials and methods

### Data

Saccharomyces cerevisiae is a beneficial yeast to analyze eukaryotes. One of the outstanding characteristics of it is that almost all bioprocesses in eukaryotes can exist in Saccharomyces cerevisiae [53]. In this study, we analyze the data of the gene interaction similarity networks of it. Costanzo and his colleagues have provided the data [48]. They have published three gene interaction similarity matrices, for essential genes, nonessential genes, and the combination of them in the global form [54, 55]. It can be helpful to perceive these two groups of genes categorized as essential and nonessential more deeply. Here, we explain the discriminator features that classify them into these two groups. First, it should be mentioned that the type of mutation generating these mutants is different. Specifically, essential and nonessential genes are mutated through temperature-sensitive and deletion mutations, respectively. Topologically, they are connected denser compared to the nonessential ones. Thus, the essential genes are considered network hubs. Moreover, in the network, essential genes show a stronger functional connection. Besides, by evaluating the predictive power, essential gene interaction profiles provide higher-accuracy gene function predictions for biological processes. At last, the biological processes specifically detected in the essential gene similarity network are cell polarity, protein degradation, and ribosomal RNA processing. Whereas, in the nonessential gene similarity network, mitochondrial and peroxisomal functions were identified [48].

The data file analyzed concerning these genes during the current study is available at http://boonelab.ccbr.utoronto.ca/supplement/costanzo2016/. We have worked with data file S3 titled "Genetic interaction profile similarity matrices". The steps taken to produce this data are as follows:

1. Based on the growth rate of the colony consisting of two specific mutated genes, the genetic interaction score (epsilon) between them has been obtained.

2. A genetic interaction profile for each gene is constructed by considering the genetic interaction score between that gene and a set of other genes in the colony.

3. The similarity between all two profiles is measured by calculating Pearson correlation coefficient ($PCC$).

The positive value in the $PCC$ matrix indicates how much those two genes are functionally similar to each other, and vice versa. Moreover, zero elements show that those two genes are not related functionally. The aforementioned procedure accomplished to obtain the $PCC$ matrices is presented in Fig 1.

For more detail, it should be pointed that the analyzed data is based on a subset of the complete Synthetic Genetic Array analysis dataset ($SGA$) of the yeast Saccharomyces cerevisiae. The $SGA$ dataset is based on genetic interactions of nonessential deletion mutants and/or essential temperature-sensitive mutants. To derive genetic interactions quantitatively, colony size is modeled as a multiplicative combination of double mutant fitness, time, and experimental factors. Succinctly, for a double mutant, carrying mutations of genes 1 and 2, colony size $c_{12}$ can be expressed as $c_{12} = f_{12} \times t \times s_{12} \times e$, where $f_{12}$ is the double mutant fitness, $t$ is the incubation time, $s_{12}$ is the combination of all systematic factors, and $e$ is log-normally distributed random noise. The $f_{12}$ is denoted as $f_{12} = f_1 f_2 + \varepsilon_{12}$, where $f_1$ and $f_2$ describe the fitness of the two

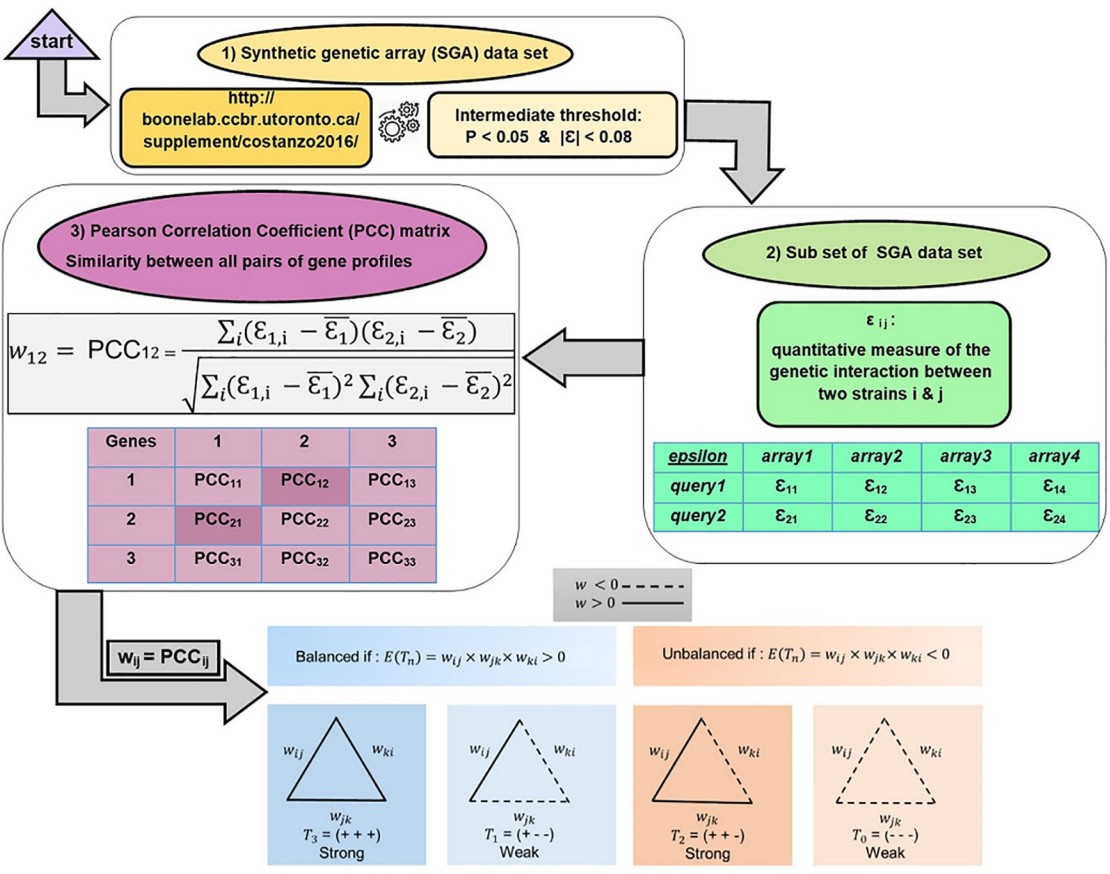

**Fig 1. Graphical abstract for the procedure of obtaining the genetic interaction similarity matrices.**

single mutants, and $\varepsilon_{12}$ (epsilon) is the quantitative measure of the genetic interaction between them. The epsilon is either positive or negative. Negative $\varepsilon$ between two mutated genes means that the combination of those two mutants causes cell death. Conversely, positive $\varepsilon$ implies that the combination of two mutated genes results in a phenotype less severe than expected. Through the $\varepsilon$, each gene has an interaction profile. In other words, that mutated gene (essential or nonessential) is crossed to a set of another mutated gene (essential or nonessential). Then, *PCC* between every two interaction profiles of genes is calculated. Indeed, each element of the *PCC* matrix, which shows the amount of similarity between every two profile interactions of genes, is between −1 and +1 [48].

## Network analysis

Prior to our main analysis, as has been carried out in the literature [1] and based on our research aims, we calculated six standard network's topological and statistical measurements, namely, mean degree ($k$), the ratio between mean of squared degrees and squared of mean degree $\left(\frac{\langle k^2 \rangle}{\langle k \rangle^2}\right)$, modularity, assortativity coefficient, average path length ($L$), and clustering coefficient ($C$). In detail, the most elementary characteristic of a network is its $k$, which tells us how many links each node has to other nodes on average. Besides, the coefficient $\langle k^2 \rangle$ holds information about the values around mean degree. However, $\langle k \rangle^2$ includes information about the

tail of degree distribution. Hence, low $\frac{\langle k^2 \rangle}{\langle k \rangle^2}$ indicates that the tail carries a higher share in the couplings. Regarding modularity, it measures the strength of a network in division into modules. Concerning assortativity (disassortativity), positive (negative) coefficient means that high-degree components often tend to be connected with similar (different) counterparts [56]. Also, $L$ declares the minimum number of edges that must be traversed to get from one node to the other [57]. Finally, the value of $C$ states the extent to which the neighbors of a node are also interconnected [58].

To compare networks with different sizes ($N$) and mean degrees ($k$) through the $N$, $k$-dependent graph measures like $C$ and $L$, a normalization technique is needed to be applied to correct the effect of $N$ and $k$. It should be highlighted while each normalization method has its advantages and disadvantages, the network type plays a key role in selecting a suitable method to mitigate the $N$, $k$-dependence of graph measures. It is worth mentioning that many empirical networks appear to have small-world characteristics [59]. To investigate if a real-world network is considered as a small-world network or not, small world index ($SW$) is utilized [59, 60]. The $SW$ is defined as the ratio between normalized $C$ and $L$. Also, $C_{rand}$ and $L_{rand}$ are those of the random network with the same number of nodes and connectivity density. Specifically, small-world networks are characterized by $C > C_{rand}$ and $L \approx L_{rand}$. Thus, a network can be a small-world network if its $SW$ index is greater than one.

$$SW = \frac{\dfrac{C}{C_{rand}}}{\dfrac{L}{L_{rand}}} \tag{1}$$

Since the values of the aforementioned indicators in a small-world network are between those of a lattice and a random network, one may express the normalized indicators as a fraction of the range of the possible obtainable values. In other words, representing normalized indicator like $\tilde{C}$ ($\tilde{L}$) as a ratio of the range of possible obtainable values declines the sensitivity to differences in $N$ and $k$ [59]. Through this normalization, $C$ ($L$) is considered as the observed indicator, $C_{rand}$ ($L_{rand}$) is the value of that in the corresponding random network, and $C_{latt}$ ($L_{latt}$) shows the value of that indicator in the lattice. Specifically, the random network is constructed by shuffling the links without any changes in the number of nodes or connectivity density. Also, the ring lattice is created by the same number of nodes with the $k$ for each node while preserving the edge densities.

$$\tilde{C} = \frac{C - C_{rand}}{C_{latt} - C_{rand}} \qquad , \qquad \tilde{L} = \frac{L - L_{rand}}{L_{latt} - L_{rand}} \tag{2}$$

Finally, it should be noted that comparing the small-worldness of two networks with different $N$ and $k$ leads to misleading results. On the one hand, the value of $L$ in small-world networks is close to that of random networks. On the other hand, the value of $C$ is contrastingly close to that of lattice networks. Thus, normalization implies a bias, i.e. the normalized $SW$ is larger than its non-normalized one. Because of this, the $SW$ is also significantly affected by $N$ and $k$. Altogether, quantifying the extent to which networks display a small-world structure is a standard way to compare their small-worldness. To this aim, as Muldoon has proposed [61] the small-world propensity ($\phi$) is calculated to reflect the deviation of a network's $C$ and $L$, from both lattice and random networks constructed with the same $N$ and $k$. In the following equation, $\Delta_C$ and $\Delta_L$ show the deviation of $C$ and $L$, that are calculated as $\Delta_C = \frac{C_{latt} - C}{C_{latt} - C_{rand}}$ and $\Delta_L = \frac{L - L_{rand}}{L_{latt} - L_{rand}}$, respectively. The value of $\phi$ which is between zero and one, is close to one for

networks with high small-world characteristics, while the lower value of $\phi$ represents less small-world structure.

$$\phi = 1 - \sqrt{\frac{\Delta_C^2 + \Delta_L^2}{2}} \qquad (3)$$

Besides graph measures, further investigations regarding the existence of structure based on spectral analysis can sure be insightful. When there is no structure beyond pairwise interactions, that network can be known as a random one. In a random network, the distribution of the spectrum of eigenvalues has a semi-circular form with a body-centered around zero [62]. In a nonrandom network, there are some eigenvalues out of the bulk [63]. Also, one large eigenvalue exists that mostly has a value far from the bulk of the eigenvalues [64, 65]. This eigenvalue plays a significant role and addresses the global trend of the system.

## Structural balance theory

To go beyond the assumption that pair interactions are independent and look for triads as the shortest motif, structural balance theory (*SBT*) is applied [29]. To consider the local triads, we focus on groups with three interacting nodes in the network. There are four types of triads, including two balanced and two unbalanced ones. The idea of "The friend of my friend is also my friend [+ + +] refers to strongly balanced triad ($T_3$)". Also, the idea of "The enemy of my enemy is my friend [− − +] points to weakly balanced triad ($T_1$)". Regarding the two other types of signed triads, [+ + −] is strongly unbalanced triad ($T_2$), and [− − −] is weakly unbalanced triad ($T_0$), which give rise to frustration in the network [44]. In other words, the triad is recognized as a balanced one if the sign of the product of its links is positive; otherwise, the triad is considered as an unbalanced or frustrated one.

As counting the number of balanced and unbalanced triads prepares informative information, significant computational methods are applied to speed up accounting for the number of triads in signed and large networks [66]. Here, we mention one of them which works based on unsigned ($A(|\Sigma|)$) and signed ($A(\Sigma)$) adjacency matrices. In the unsigned adjacency matrix, if the nodes $i$ and $j$ are connected then $A(|\Sigma|)(i, j) = 1$, otherwise $A(|\Sigma|)(i, j) = 0$. In the signed adjacency matrix, if the link's sign connecting those nodes is positive then $A(\Sigma)(i, j) = 1$, and if the link's sign connecting those nodes is negative then $A(\Sigma)(i, j) = −1$. As follows, the two equations count the number of balanced ($b$) and unbalanced ($u$) triads, respectively:

$$b = \frac{1}{12}[trace(A(|\Sigma|)^3) + trace(A(\Sigma)^3)], \qquad (4)$$

$$u = \frac{1}{12}[trace(A(|\Sigma|)^3) - trace(A(\Sigma)^3)]. \qquad (5)$$

As Leskovec has proposed [3], we have created a null model to compare the empirical frequencies of triads. It is important for generating a null model to keep the exact fraction of positive (negative) signs. Specifically, each randomly chosen link connecting the two existing nodes is shuffled. Thus, the created null model represents no organization in the structure. Then, the fraction of each type of triad in the shuffled network ($p_0(T_i)$) is calculated. The triad $i$ is overrepresented if the related fraction in the original network ($p(T_i)$) be more than that of in the shuffled one; otherwise, it is underrepresented. Next, the value of surprise ($s(T_i)$) is calculated which is the number of standard deviations by which the actual number of triad $i$ differs from its expected number under the null model. Within the function of ($s(T_i)$), $T_i$ is the number of triad $i$, $E[T_i]$ is the expected number of triad $i$ calculated as $E[T_i] = \Delta p_0(T_i)$, and $\Delta$ is

the total number of triads calculated as $\Delta = trace(A(|\Sigma|)^3$. To eliminate the effect of size in both networks, after calculating the $s(T_i)$ function, it is divided into $\sqrt{\Delta}$.

$$s(T_i) = \frac{T_i - E[T_i]}{\sqrt{\Delta p_0(T_i)(1 - p_0(T_i))}} \tag{6}$$

It has been stated that a balanced network is a network consisting of all positive triads [8]. While the possibility of possessing a real-world network containing all positive signed triads (positive product of their sides) is close to zero. Thus, a common approach is to measure the degree of balance of a signed network. To this aim, the concept of balance enables us to determine an energy landscape for such networks. Energy describes how much a network is structurally balanced [21, 67]. In a weighted network, the network energy ($E$) is obtained by the negative summation of the products of the triads' links ($w_{ij} w_{jk} w_{ki}$) divided by weighted sum of all triads' energies ($\Delta_w$) which is calculated as $\Delta_w = \sum_{i<j<k}^N |w_{ij} w_{jk} w_{ki}|$. For a balanced triad, the product of its weighted links is a positive number, whereas for an unbalanced one this product is negative. If $E = -1$, then we have a fully balanced network. But if it equals $+1$, then we have an unbalanced network. Consequently, in real-world networks, the energy of triads is between $-1$ and $+1$. According to *SBT*'s suggestion, a network evolves towards the minimum level of tension [67].

$$E = -\frac{1}{\Delta_w} \sum_{i<j<k}^N w_{ij} w_{jk} w_{ki} \tag{7}$$

The energy landscape introduced above considers the triads individually and does not designate how they are organized in the network globally. Put differently, after calculating the energy of each triad, we aim to investigate how they are connected through one shared link. The following questions are our concerns in this regard: Do triads form a module, or are isolated? Does a triad with a high (low) energy value tend to be connected with triads of different energies? What types of triads a specific triad with a defined energy value is connected to, and with what energy value? To answer these questions, the energy-energy mixing pattern is plotted. To be more specific, through moving on sorted spectrums of energy of two specific types of triads, the number of triads that have a common link is counted. This calculation is repeated for all pair types of triads. Indeed, this pattern shows if particular types of triads are packed together and form a kind of module. Also, it figures out if triads represent a heterogeneous (homogeneous) form of connections. Moreover, it clarifies if a triad with a high (low) energy value tends to be connected with triads of different (similar) energies.

## Walk-based measure of balance and detecting lack of balance

*SBT* gives specific information to understand the structural balance of signed networks but is biased. Through triads, our analysis recognizes the frustration on the shortest possible cycle, but it overlooks to consider the unbalancing correlated with longer-range cycles [33]. To extend our analysis by considering cycles with all possible lengths, it should be mentioned that the balance or unbalancing of each cycle is related to the multiplication of the signs of its links. If the sign of the product is positive, or the number of negative links in the cycle is even, it is a balanced cycle. Therefore, if all cycles in a network have a positive sign, we can consider the signed network as a balanced one [44–46]. But the fact is that the probability of having a real-world network containing all cycles with a positive sign is close to zero. As Estrada proposed in [47], the walk-balance index ($K$) is used to quantify how close to balance an unbalanced network is. Specifically, walks with all lengths are considered concerning assigning more weights

to the shorter ones, which is logical [47]. This method relates a hypothetical equilibrium between the real-world signed network and its underlying unsigned version. In $K$, $A(\Sigma)$ and $A(|\Sigma|)$ are signed and unsigned adjacency matrices, respectively. Elements in $A(\Sigma)$ are + 1 when the interaction matrix values are more than zero. Also, if the interaction matrix values are less than zero the elements in $A(\Sigma)$ are −1. In the unsigned adjacency matrix $A(|\Sigma|)$, if the elements in the interaction matrix are nonzero, the elements of $A(|\Sigma|)$ are 1. Another index proposed by Estrada measures the extent of the lack of balance in the network ($U$), as follows [47]:

$$K = \frac{trace(exp(A(\Sigma)))}{trace(exp(A(|\Sigma|)))}, \tag{8}$$

$$U = \frac{1 - K}{1 + K}. \tag{9}$$

The value of $K$ as the density of the balanced walks with all lengths in the network is between zero and one. To be specific, when the expansion of the $exp(A(\Sigma))$ in $K$ is opened, among walks with all possible lengths, there can be some negative terms in the nominator, although, in the denominator, all terms of expansion of the $exp(A(|\Sigma|))$ are positive. Thus, if all present walks are positive (a balanced network), then this index calculating the amount of balance of the network meets its maximum value, which is one. Additionally, $U$ calculating the amount of unbalance would have its minimum value, which is zero. At last, the participation of each node in the balance of the network can be calculated by the degree of balance of a given node $i$ as $K_i$ [47]. According to the following equation, $K_i$ flows between zero and one. Thus, the term "highest degree balance" is assigned to the nodes with $K_i = 1$ that participate only in the walks with an even number of negative links. That is, all walks they are joining in are balanced.

$$K_i = \frac{exp(A(\Sigma))_{ii}}{exp(A(|\Sigma|))_{ii}} \tag{10}$$

## Results

Based on our main research questions, six standard, and informative network's indicators, i.e., mean degree ($k$), the ratio between mean of squared degrees and squared of mean degree $\left(\frac{\langle k^2 \rangle}{\langle k \rangle^2}\right)$, modularity, assortativity coefficient, average path length ($L$), and clustering coefficient ($C$) are calculated. Specifically, through computing $C$ we observe networks' tendency to form triads, which are the basic building blocks in the balance theory framework. As well, modularity provides information on the networks' communities, which is a very crucial feature in gene network studies. Also, if $k$ of networks, besides their sizes ($N$), be different, to compare those networks, a normalization technique which is related to the topology of networks should be selected to normalize $N$, $k$-dependent network's indicators like $C$ and $L$. Since most real-world networks have small-world topology, according to Eq (1), the small-world index ($SW$) in our networks is calculated. The result indicates that the $SW$ in both essential and nonessential gene networks as the same as in small-world networks is greater than one, which is 1.0866 and 1.2402, respectively. Thus, according to what small-world structure implies (the values of $C$ and $L$ are between their values of lattice and random versions), $C$ and $L$ through Eq (2) within a range of possible values are normalized.

Indicators in both essential and nonessential gene networks are compared in Fig 2. Despite the segregation among the measurements, there exist some similarities. As shown in Fig 2, the $k$ in the nonessential gene network is higher compared to the essential network. Besides, in both

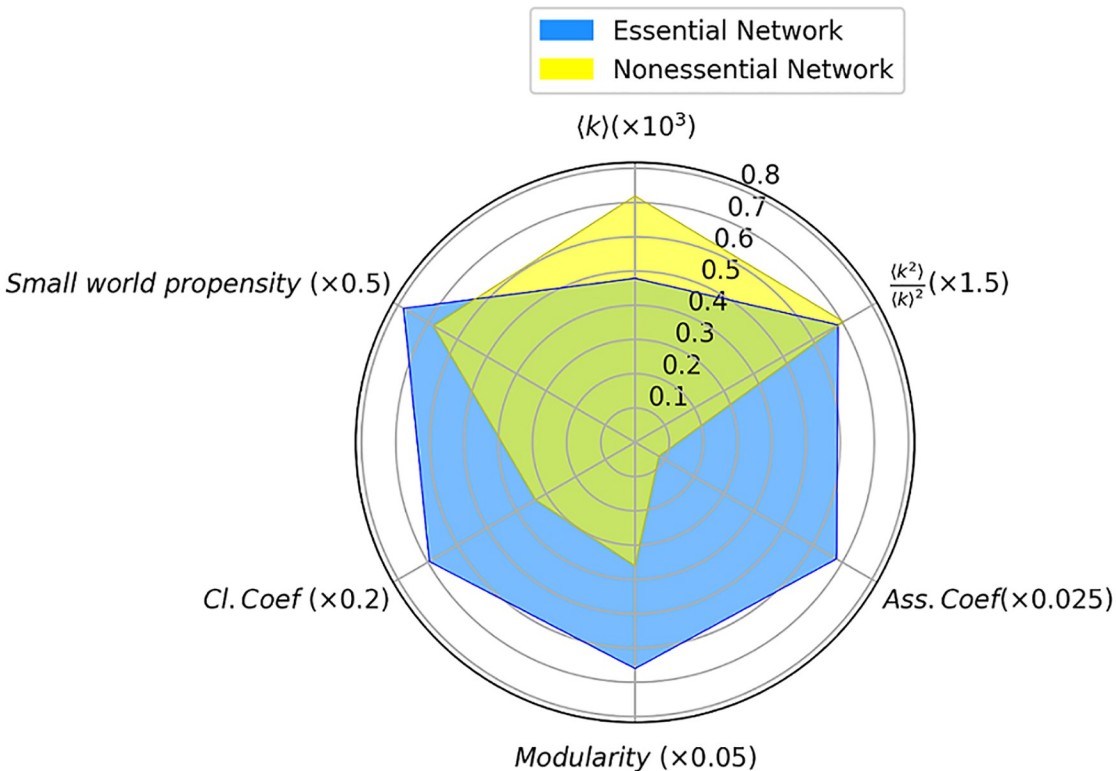

**Fig 2. The radar plot shows six standard network's indicators in both essential and nonessential gene networks.** Mean degree, the ratio between mean of squared degrees and squared of mean degree, modularity, assortativity coefficient, normalized clustering coefficient, and small-world propensity. The radar plot for the essential gene network is plotted in blue and for the nonessential gene network in yellow.

networks, the ratio between mean squared degrees and squared of mean degree is close to one. This implies that neither nodes with high degrees nor those with a medium degree are significantly dominant over the other one. In addition, the value of the modularity in the essential network is more than that of in the nonessential network. The higher value of this indicator in the essential gene network than the nonessential one indicates the higher tendency to be clustered into multiple sets of strongly interacting parts. Moreover, as it has been illustrated in Table 1, the assortativity coefficient in both networks is negative but so close to zero, i.e., both networks show weak disassortative behavior. However, the magnitude of disassortativity is one order

**Table 1. Network's indicators.**

|  | Essential | Nonessential |
|---|---|---|
| **Mean degree (k)** | 478.890 | 718.957 |
| $\frac{\langle k^2 \rangle}{\langle k \rangle^2}$ | 1.0278 | 1.0560 |
| **Modularity** | 0.033 | 0.018 |
| **Assortativity** | −0.017 | −0.002 |
| **Normalized clustering.coef $(\tilde{C})$** | 0.139 | 0.067 |
| **Small world propensity $(\phi)$** | 0.391 | 0.340 |

Mean degree, the ratio between mean of squared degrees and squared of mean degree, modularity, assortativity coefficient, normalized clustering coefficient, and small-world propensity for both essential and nonessential gene networks.

higher in the essential network. In the radar plot (Fig 2), the absolute values of assortativity coefficients are demonstrated. Additionally, the values of normalized $L$, in both networks are close to zero, which shows that these networks are densely connected, and there is a very small difference between values of observed $L$ with those of shuffled versions. As well, the tendency in forming clusters is defined by the normalized $C$ which is higher in the essential network. At last, because of the size dependence of $SW$, through Eq (3) the small-worldness propensity ($\phi$) of networks is calculated to understand the extent of this characteristic in our networks. The value of $\phi$, for the essential gene network, is larger compared to the nonessential network.

Then, we have investigated the existence of clusters in the construction of the essential and nonessential gene networks. Within groups, genes cooperate to annotate a common bioprocess efficiently. Clusters in both essential and nonessential gene networks are illustrated through cluster maps (Fig 3). It can be seen that the essential network has stronger structural modules which are in line with the previous result which stated that the essential network is more modular than the nonessential network. In other words, although the clusters exist in both networks, the structure in the essential gene network (Fig 3A) is highly stronger than the nonessential network (Fig 3B). This also confirms our previous study, where we observed a significant difference between the distributions of eigenvalues in the original matrices and those of the shuffled networks [52]. To be specific, some of the eigenvalues in the original networks are not limited to the narrow bulk of the eigenvalues in the shuffled matrices. Thus, it can be confidently concluded that the structure of the gene interaction networks is far from random.

After studying the clusters, the structural balance in gene interaction networks to study the structure beyond pairwise interactions is analyzed. To this aim, as the first step, the size, the percentage of positive and negative links, and the total number of triads in both networks are prepared (Table 2). In the following, the two equations Eqs (4) and (5) are utilized to count balanced ($b$) and unbalanced ($u$) triads. Then, to compare the dominance of balanced or unbalanced triads in our networks, we have applied the method proposed by Leskovec et al. [3]. According to this method, if the fraction of balanced (unbalanced) triads in the original network is higher than the shuffled one, it will overrepresent, and vise versa. Through this method, the fraction of the triad $T_i$ in the original network is considered as $p(T_i)$ and in the shuffled network as $p_0(T_i)$. Moreover, they have proposed the concept of surprise as Eq (6), $s$

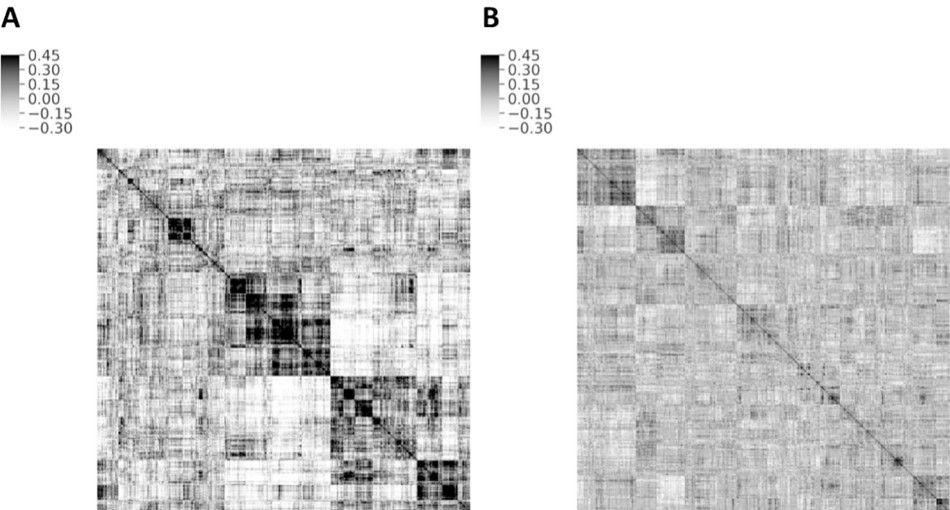

**Fig 3. The cluster map of two essential and nonessential gene networks.** A: Cluster map of essential gene network, B: Cluster map of nonessential gene network.

**Table 2. Dataset statistics.**

|  | Essential | Nonessential |
|---|---|---|
| **Nodes** | 1, 040 | 4, 430 |
| **Edges** | 249, 023 | 1, 592, 490 |
| **+Edges** | 50.1% | 63.5% |
| **−Edges** | 49.9% | 36.4% |
| $\frac{\text{Edges}}{\binom{N}{2}}$ | 0.461 | 0.162 |
| **Triads** | 20, 310, 741 | 81, 470, 554 |
| $\frac{\text{Triads}}{\binom{N}{3}}$ | 0.109 | 0.006 |

Number of nodes, edges, triads in both essential and nonessential gene networks with threshold $w_{ij} < |0.05|$.

($T_i$), to understand how significant these over (under) representations are. Due to the size of the networks, $s(T_i)$ has a significant order of tens. The results indicate that balanced triads are overrepresented in both essential and nonessential gene interaction networks. On the contrary, unbalanced triads are underrepresented compared to their shuffled versions. These results are presented in Table 3.

After analyzing the frequency of triads, we have examined the energy distribution of different types of triads. Thus, we have calculated the energy of triads by Eq (7). Then, the energy distributions of strongly balanced triads ($T_3$) in Fig 4A, weakly balanced triads ($T_1$) in Fig 4B, strongly unbalanced triads ($T_2$) in Fig 4C, and weakly unbalanced triads ($T_0$) in Fig 4D for both original networks, in comparison with their shuffled versions, are presented. Results indicate: (1) All types of triads, in both essential and nonessential networks, have many triads with small values of energies. (2) In the essential gene network, the largest amount of triads' energy is for the $T_1$ triads, and in the nonessential gene network, the $T_3$ triads have the largest value of energy (Fig 4E). (3) In both gene networks, the bar levels of the average energy of balanced triads are higher than those of shuffled ones. However, on the contrary, the bar levels of the average energy of unbalanced triads are lower than those of shuffled ones. (4) As Fig 4F, in the essential gene network, the relative frequency of the balanced triad $T_1$ is individually equal to the relative frequency of the other three types of triads.

**Table 3. Number and probability of balanced and unbalanced triads in the original networks compared to the null model.**

| Essential gene network | $|\mathbf{T_i}|$ | $\mathbf{p(T_i)}$ | $\mathbf{p_0(T)}$ | $\mathbf{s(T_i)}$ | $\frac{s(T_i)}{\sqrt{\Delta}}$ |
|---|---|---|---|---|---|
| *Strongly balanced* ($T_3$) | 3, 670, 948 | 0.180 | 0.124 | 764.0 | 0.2 |
| *Weakly balanced* ($T_1$) | 10, 362, 180 | 0.510 | 0.375 | 1, 255.1 | 0.3 |
| *Strongly unbalanced* ($T_2$) | 4, 421, 666 | 0.217 | 0.374 | −1, 461.1 | −0.3 |
| *Weakly unbalanced* ($T_0$) | 1, 855, 947 | 0.091 | 0.125 | −462.0 | −0.1 |
| **Nonessential gene network** | $|\mathbf{T_i}|$ | $\mathbf{p(T_i)}$ | $\mathbf{p_0(T)}$ | $\mathbf{s(T_i)}$ | $\frac{s(T_i)}{\sqrt{\Delta}}$ |
| *Strongly balanced* ($T_3$) | 30, 868, 604 | 0.378 | 0.256 | 2, 531.1 | 0.3 |
| *Weakly balanced* ($T_1$) | 32, 704, 022 | 0.401 | 0.253 | 3, 071.6 | 0.3 |
| *Strongly unbalanced* ($T_2$) | 16, 028, 365 | 0.196 | 0.441 | −4, 452.8 | −0.5 |
| *Weakly unbalanced* ($T_0$) | 1, 869, 563 | 0.022 | 0.048 | −1, 071.7 | −0.1 |

$|T_i|$ = the total number of triads of type $i$; $p(T_i)$ = the fraction of $T_i$; $p_0(T_i)$ = the fraction of $T_i$ in the null model; $s(T_i)$ = the amount of surprise, i.e., the number of standard deviations by which the actual number of $T_i$ differs from its expected number under the null model; and $\Delta$ = the total number of triads.

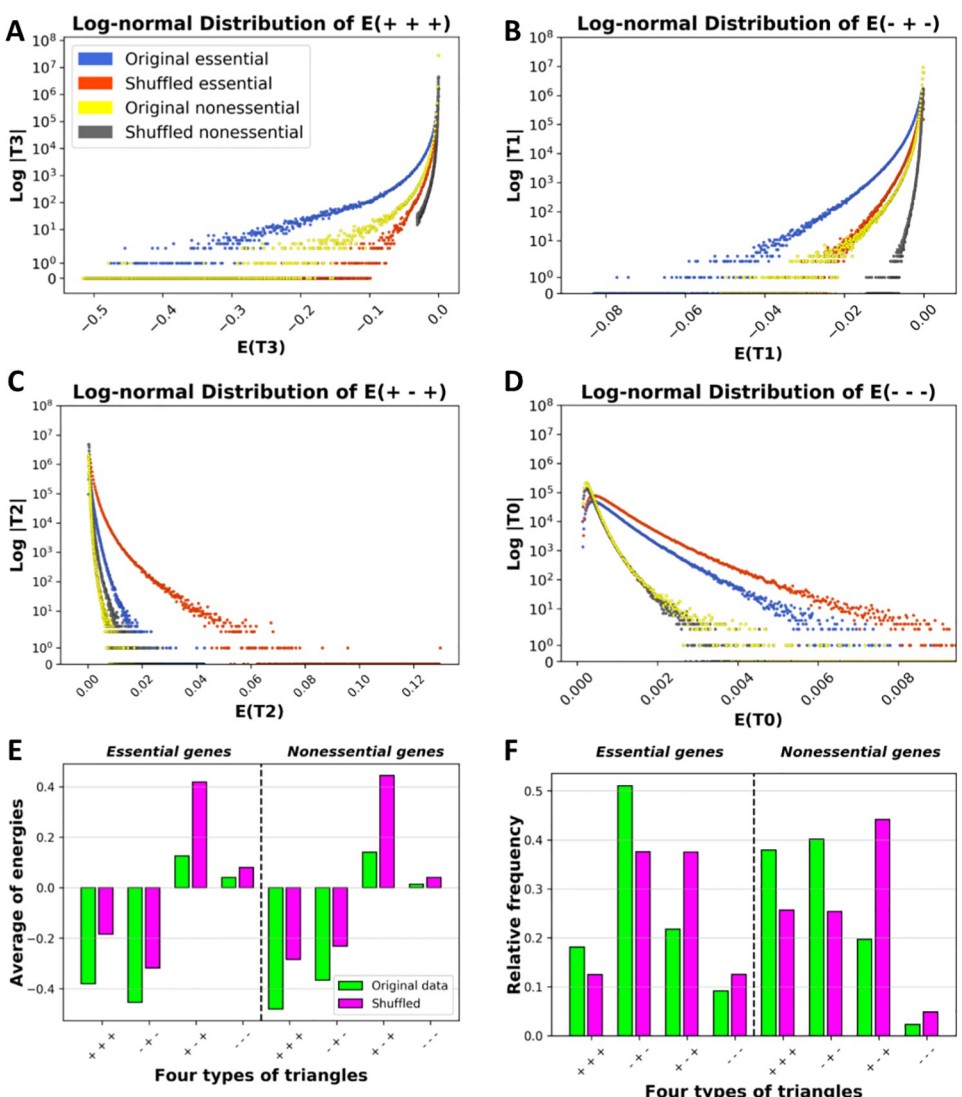

**Fig 4. Energy distributions, average energy, relative frequency for all four types of triads.** A: Energy distribution for strongly balanced triads, B: Energy distribution for weakly balanced triads, C: Energy distribution for strongly unbalanced triads, D: Energy distribution for weakly unbalanced triads. (The energy distribution of triads for original essential gene network and its shuffled network are plotted in blue and red, respectively. The energy distribution of triads for original nonessential gene network and its shuffled network are plotted in yellow and gray, respectively). E: From left to right, the average energy for essential gene network and nonessential gene network. F: From left to right, the relative frequency for essential gene network and nonessential gene network (Green bars for original networks and purple ones for shuffled networks).

Here, we intend to understand how triads are globally organized in the network. To address this aim, the energy-energy mixing pattern in the logarithmic scale has been plotted in Fig 5. By using the logarithmic scale, there is a magnification between the elements with small amounts. Specifically, our goal is to enrich our analysis by studying patterns of the connection between triads. Results reveal that there are fewer connected triads compared to isolated ones overall. Moreover, $T_1$ triads are more connected to each other compared to other types. Furthermore, triads with low absolute energy values have more tendency to be connected

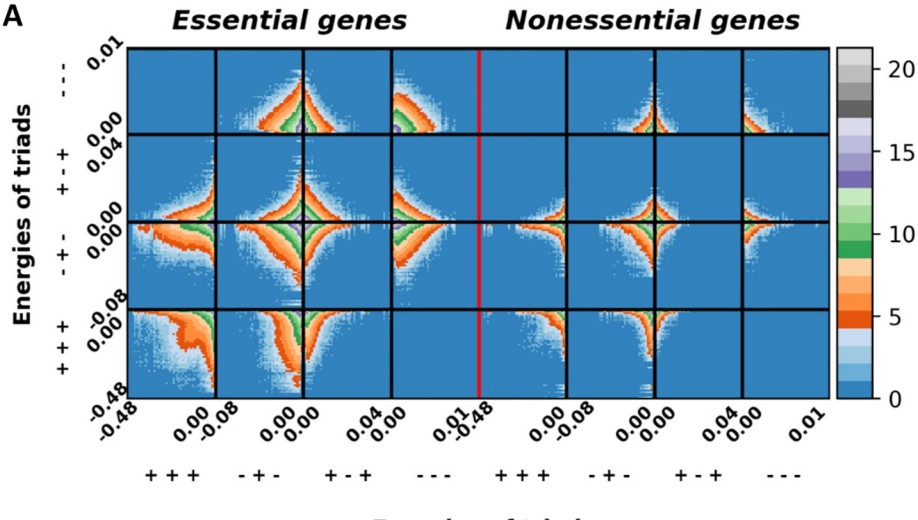

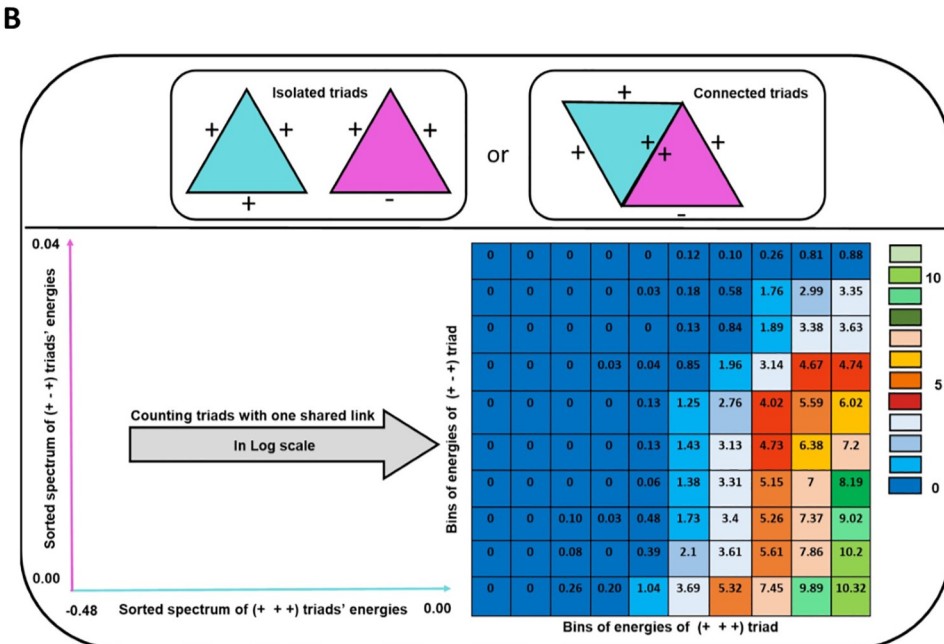

**Fig 5. The pattern of connection between triads through one shared link in Log scale.** A: All types of pair connected triads (From left to right, essential gene network and nonessential gene network), B: An overview of creating connection between triads for one square ($T_3$ [+ + +] and $T_2$ [+ - +] triads).

compared to high energy triads. While this pattern holds for both essential and nonessential gene networks, the essential network has more triads with the shared link. To clarify Fig 5A more clearly, the following steps are taken to plot each square in Fig 5B:

1. The spectrums of energy of two specific types of triads are sorted.

2. Through moving on the energy axes, the number of triads that have a common link is counted and saved in a matrix in the Log scale.

**Table 4. Walk-balance index for all cycles (K), percentage of the lack of balance (U).**

|  | Essential | Nonessential |
|---|---|---|
| $K_{\text{original network}}$ | 0.195 | 0.988 |
| $K_{\text{shuffled}}$ | 0.000 | 0.131 |
| $U_{\text{original network}}(\%)$ | 67.238 | 0.575 |
| $U_{\text{shuffled}}(\%)$ | 99.999 | 76.749 |

For the original and shuffled of essential and nonessential gene networks with threshold $w_{ij} < |0.2|$.

3. The previous steps are repeated for all pair types of triads.

4. All 16 squares in 4 rows and 4 columns are merged.

Now, by considering walks with all possible lengths, we extend our analysis. To this aim, the quantity of balance or unbalancing through these walks is measured. Indeed, we employed two indices introduced in [47] by Estrada not to limit ourselves only to triads as the shortest cycle. One of the two indices is the walk-balance index ($K$) which calculates by Eq (8) the amount of how close to balance an unbalanced network is. Another index represents the extent of the shortage of balance ($U$) in a given signed interaction network by Eq (9). In Table 4, the values of the $K$ in both essential and nonessential gene networks have been presented. For each index in both networks, there is a leading difference between the value of the original and that of the shuffled matrix. The fact is that in an unbalanced network, for example, a random network that holds no structural information, the $K$ would have the lowest possible value i.e., a value close to zero. Also, $U$ would have the highest possible value, that is, a value close to one. As the result indicates, by considering all walks, both essential and nonessential gene networks are more close to balance rather than their corresponding shuffled versions. Besides, in the nonessential gene network, $K$ is higher than the essential gene network. Moreover, the $U$ in the essential gene network is much more than the nonessential gene network. Furthermore, there is an index that characterizes the degree of balance for a given node $i$ as ($K_i$) by Eq (10). In supplementary, a table is prepared to represent the classification of genes with the highest degree of balance ($K_i = 1$) in terms of biological processes they annotate.

## Discussion

We analyzed gene interactions in the weighted, undirected, and signed networks of yeast Saccharomyces cerevisiae. The pre-processed dataset used includes two matrices, namely, essential and nonessential gene interaction networks. Here, we explored these two gene networks beyond pairwise interactions in the context of structural balance theory ($SBT$). The following results have been concluded accordingly: We have discovered that in both essential and nonessential gene networks balanced triads are overrepresented while unbalanced triads are underrepresented. Interestingly, this finding is in agreement with Heider's balance theory. To be specific, our results empirically support the strong notion of structural balance theory (Table 2). This is while in some social networks, the weak formulation of structural balance has been reported as well.

Additionally, we have observed $T_1$ and $T_0$ triads in both gene networks with more average energy and higher relative frequency in the essential network. This can be interpreted from the perspective of $SBT$ in which the presence of $T_1$ and $T_0$ triads in the organization of a network is related to having a higher degree of modularity. In other words, to have $T_1$ or $T_0$ triads in the

stable state of a network indicates that densely connected modules are also connected to others through negative links. This result corresponds to the presence of specialized clusters in the gene interaction network which has also been reflected in the energy-energy mixing pattern between the triads with one common link. It is worth mentioning that this pattern is more significant in the essential network as genes in this network are more densely interconnected.

Moreover, we have noted that although energies of the essential and nonessential networks are not significantly different from each other, the underlying triads' distributions that led to these final energies are not similar. As mentioned earlier, the average energy and the relative frequency of unbalanced triads $T_0$ are higher in the essential gene network compared to the nonessential network. Thus, they are more likely to experience different possible states. Therefore, it can be concluded that unbalanced triads $T_0$ are providing the essential gene networks with the necessary structure that is needed to contain dynamism which is crucial for vital biological mechanisms. This is while for nonessential genes with less unbalanced triads $T_0$, the likelihood of being trapped in a local minimum is higher.

Finally, to extend our analysis we have calculated two indices by considering the walks with all possible lengths. Namely, the quantification of how close to balance an unbalanced network is, and the extent to which a given signed network lacks balance by considering longer-range cycles. Results surprisingly suggest that when all length walks are taken into account, both essential and nonessential gene networks are more balanced than expected from a random allocation of the signs to the links. In other words, both essential and nonessential gene networks, besides balanced triads, respect balanced long-range interactions. Moreover, the nonessential gene network is more balanced and stable than the essential network. As mentioned earlier, the combination of both essential and nonessential interactions constructs the global gene network as a whole. For this network, we have proposed a list of genes in terms of biological processes they annotate in the S1 File that have the highest degree of balance. Thus, our finding highlights the genes that are structural of note, regarding which further biological analysis seems to be very much valuable.

## Supporting information

**S1 File.**
(PDF)

## Acknowledgments

N.A. would like to express her appreciation to Z. Moradimanesh and S. Salekzamankhani for constructive comments that improved the manuscript.

## Author Contributions

**Conceptualization:** Nastaran Allahyari, Ali Hosseiny, G. R. Jafari.

**Formal analysis:** Nastaran Allahyari, Amir Kargaran, Ali Hosseiny.

**Investigation:** Nastaran Allahyari, G. R. Jafari.

**Methodology:** Nastaran Allahyari, Ali Hosseiny.

**Project administration:** G. R. Jafari.

**Resources:** Nastaran Allahyari, Amir Kargaran.

**Software:** Nastaran Allahyari, Amir Kargaran.

**Supervision:** Ali Hosseiny, G. R. Jafari.

**Visualization:** Nastaran Allahyari.

**Writing – original draft:** Nastaran Allahyari.

**Writing – review & editing:** Nastaran Allahyari.

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
