## [Decision Letter · Decision Letter 0]

16 Nov 2021

PONE-D-21-31371The structure balance of gene-gene networks beyond pairwise interactionsPLOS ONE

Dear Dr. Jafari,

Thank you for submitting your manuscript to PLOS ONE. After careful consideration, we feel that it has merit but does not fully meet PLOS ONE’s publication criteria as it currently stands. Therefore, we invite you to submit a revised version of the manuscript that addresses all the points raised by the reviewers. Most of them simply require better explanations, but anyway they are important for the understanding of your work.

We look forward to receiving your revised manuscript.

Kind regards,

Sergio Gómez

Academic Editor

PLOS ONE

Journal Requirements:

Reviewers' comments:

Reviewer's Responses to Questions

**Comments to the Author**

1. Is the manuscript technically sound, and do the data support the conclusions?

Reviewer #1: Yes

Reviewer #2: Partly

2. Has the statistical analysis been performed appropriately and rigorously? 

Reviewer #1: N/A

Reviewer #2: Yes

3. Have the authors made all data underlying the findings in their manuscript fully available?

Reviewer #1: Yes

Reviewer #2: Yes

4. Is the manuscript presented in an intelligible fashion and written in standard English?

Reviewer #1: Yes

Reviewer #2: Yes

5. Review Comments to the Author

Reviewer #1: The authors analyse the structure of a gene interaction networks of the yeast Saccharomyces cerevisiae through the lens of structural balance theory. In this sense, they provide evidence about the structural differences between the essential network and the nonessential one and, for each of them, the difference with respect to their shuffled counterpart. In this way, they highlight the strongly non random nature of the gene networks –as quantified through several measures of balance–, while reporting some patterns that may be of potential interest for further research.

There is only one, major problem in the present analysis for which I could not find a way out. It regards the way in which the distributions of energy for the four type of triads are constructed (figure 4). Indeed, to each type of triad is assigned a certain energy, given by the product of the signs of the three links forming the triad, multiplied by -1. Accordingly, the energy is -1 for the balanced triads (T_1 and T_3) and 1 for the unbalanced ones (T_0 and T_2). (Then, as stated by Eq.4, the normalised sum over all the triads provides the energy of the entire network.) Nonetheless, in figure 4 (and 5, consequently), those energies take values on a continuous range, whereas only the discrete values of -1 and 1 should be allowed. Given that, it is really unclear from where those distributions come from. As a consequence, the same holds for the energy-energy correlations reported in figure 5. Given the important weight that such findings have on their work overall, the authors should made this point very clear.

The manuscript is well written. I can recommend it for publication once the authors will have addressed that pinpointed issue.

Please, also note the typos listed below:

- remove question mark in line 11

- change "Leskovek" to "Leskovec" in line 132

Reviewer #2: The paper computes balance for (1) triangles (3-cycles) and (2) all-length walks in gene-gene interaction networks for essential and nonessential genes for the yeast Saccharomyces cerevisiae. Overall, the paper finds that more balance is present in the nonessential genes network than the essential genes in terms of both triad- and all-walks balance. The paper is well-structured and the balance analyses are thorough.

1. Operationalization: It is still unclear what energy-energy mixing means here, specifically in terms of defining what constitute a 'positive' and 'negative' link between two genes. I suggest the authors to set a stronger background that explains how they attribute signs to edges.

2. Operationalization: It would be useful to explain further what constitute an essential vs. nonessential gene; would be helpful to readers from other disciplines.

3. Operationalization: It is still unclear how you define "high impact" of significant genes (page 8)? Do you consider genes that sit on many 'balanced' walks, or are there any measures you used to operationalize 'impact'?

4. Findings: You reported mean degree, the ratio between mean of squared degrees and squared of mean degree, modularity, assortativity coefficient, average path length, and clustering coefficient as indicators of a network's topology. They are useful but not the only indicators of topology; explanations to why these particular measurements are used would be useful.

5. Findings: In table 4, there's an index value of 0 for genes-K shuffled cell. Can you explain the meaning for index of 0 here?

6. Findings: Given the unequal sizes of the genes vs. nonessential genes network, with the nonessential genes network being much bigger, that could explain the higher 'likelihood' for finding balanced configurations for this particular network. Do you consider controlling for the size of the networks, and/or normalization techniques to mitigate the sampling/size difference here?

6. PLOS authors have the option to publish the peer review history of their article (what does this mean?). If published, this will include your full peer review and any attached files.

Reviewer #1: No

Reviewer #2: No

---

## [Author Response · Author response to Decision Letter 0]

22 Dec 2021

Respected Editor,

The authors would like to express their gratitude to you and the referees for the careful and thorough reading of our manuscript titled "The structure balance of gene-gene networks beyond pairwise interactions" and for providing fruitful comments. We found all comments very crucial, and we have modified our manuscript accordingly as follows:

1) The following sections have been modified according to the PLOS ONE style templates: (A) Affiliations and the corresponding authorship, (B) The order of sections and subsections so that the title of paragraphs in the method section is replaced with subsections, (C) All figures and tables are modified.

2) The first paragraph of the "data" subsection of the method section is enriched to describe the essential and nonessential genes in more detail. Besides, a new paragraph is added to have a stronger background in the data preparation steps.

3) In the "structural balance theory" subsection of the method section, Eq (7) and two sentences within its previous paragraph are modified by replacing S with W to prevent possible misunderstanding of using signs instead of weights. Also, ∆ is replaced with ∆ w to apply the weighted sum of all triads’ energies instead of the binary summation. Hence, Fig 4E is modified. Moreover, to have better consistency in terms of notations, we have updated the terms "adjacency matrix, A" and "connectivity matrix, G" in Eq (4) and (5) to "signed adjacency, A(Σ)" and "unsigned adjacency A(|Σ|)" according to Eq (8), respectively.

4) To the last paragraph of the "walk-based measure of balance and detecting lack of balance" subsection of the method section, a few sentences are added to better clarify the notion of walk balance index (K) and degree of balance index of a given node (K i ), respectively.

5) Three new paragraphs to the "network analysis" subsection of the method section are added to explain how we are allowed based on the network topology to compare size-dependent indicators of networks. To cite the applied method, we have also added three new references as numbers 59-61 to the "network analysis" subsection. Besides, two paragraphs at the beginning of the result section, Table 2, and Fig 2 are modified.

6) In the result section, a new subfigure is added as Fig 5B, along with new explanations to better clarify Fig 5A.

7) The supplementary file is updated by classifying the reported genes with the highest degree of balance in terms of the biological processes that are annotated.

Please find our attached point-by-point response to the reviewers concerns. All changes are marked red in the revised manuscript with track changes, and pages numbers that are provided in this response are according to this file as well. We highly appreciate your valuable time in considering the revised version of our manuscript.

Sincerely,

G. R. Jafari

Full professor

Department of Physics, Shahid Beheshti University, Evin, Tehran, Iran

Institute of Information Technology and Data Science, Irkutsk National Research Technical University, Lermontova, Irkutsk,

Russia

Email: g_jafari@sbu.ac.ir, gjafari@gmail.com

#Reviewer 1

1. There is only one, a major problem in the present analysis for which I could not find a way out. It regards the way in which the distributions of energy for the four types of triads are constructed (figure 4). Indeed, each type of triad is assigned certain energy, given by the product of the signs of the three links forming the triad, multiplied by -1. Accordingly, the energy is -1 for the balanced triads (T 1 and T 3 ) and 1 for the unbalanced ones (T 0 and T 2 ). (Then, as stated by Eq.4, the normalized sum over all the triads provides the energy of the entire network.) Nonetheless, in

figure 4 (and 5, consequently), those energies take values on a continuous range, whereas only the discrete values of -1 and 1 should be allowed. Given that, it is really unclear where those distributions come from. As a consequence, the same holds for the energy-energy correlations reported in figure 5. Given the important weight that such findings have

on their work overall, the authors should make this point very clear.

Response: Thanks to the respected reviewer for this fruitful comment. It should be mentioned that in Eq (4), the energy of the network is computed by the product of −1 into the “weights” of the three links forming the triad (not just the signs), divided by the weighted sum of all triads’ energies (∆ w ) in the network. That is, “w i j , w jk , w ki ” as the links’ weights of the “i jk” triad are multiplying to give its energy. To correct this in the manuscript, we replaced S in Eq (1) with W according to Eq (2) to avoid misunderstanding the sign instead of weight. Besides, we replaced ∆ with ∆ w to apply the weighted sum of all triads’ energies instead of the binary summation, as follows:

E = − Σ s ij s jk s ki / ∆ (s = ±1) (1)

 to 

E = − Σ w ij w jk w ki / ∆ w (−1 ≤ w ≤ +1). (2)

Therefore, the distribution of energy for balanced triads T 3 and T 1 (Fig 4A and B), as well as for unbalanced triads T 2 and T 0 (Fig 4C and D) is calculated by multiplying −1 to the weight of links of each three interconnected genes to each other. Consequently, in Fig 5, energies can take values on a continuous range between −1 and +1. It is worth mentioning, in the standard balance theory, links are unweighted and characterized only by positive and negative signs. This is while in real- world networks, weights of links are as crucial as signs 1 . A part of the matrix we worked on is here. As can be seen, it is a weighted and signed network with links weights between −0.4 to +0.8 (Fig 1). 

Figure 1. A part of the studied matrix as an example for further clarification 2 .

To make this indispensable point clearer in the manuscript, we add two sentences for a more precise explanation in the method section, on page 6, lines 229-234. Furthermore, Eq (4) (in the revised manuscript Eq (7)) is corrected by replacing S with W, and by using the weighted sum of all triads’ energies (∆ w ) instead of binary summation (∆). Hence, we modified Fig 4E. We would like to greatly thank the reviewer for the careful, and insightful review and thoughtful comment.

2. Please, also note the typos listed below:

-remove question mark in line 11

-change "Leskovek" to "Leskovec" in line 132

Response: Thanks to the reviewer for this comment. Both typos are corrected in the revised manuscript.

#Reviewer 2

1. Operationalization: It is still unclear what energy-energy mixing means here, specifically in terms of defining what constitutes a ’positive’ and ’negative’ link between two genes. I suggest the authors set a stronger background that explains how they attribute signs to edges. 

Response: Thanks to the respected reviewer for this fruitful comment. Besides calculating triads’ energies (Fig 4), we aimed to understand how they are organized globally in the network by using energy-energy mixing analysis. Specifically, in the energy-energy mixing (Fig 5), our goal is to enrich our analysis through studying nontrivial patterns according to which triads are connected. In other words, after considering each triad individually, we intended to extract information about the connections between them. To be more specific, the following questions were our concerns in this regard: Do triads form a module, or are isolated? What types of triads a specific triad with a defined energy value is connected to, and with what energy value? Does a triad with a high (low) energy value tend to be connected with triads of different energies? Results reveal that there are fewer connected triads compared to isolated ones overall. Moreover, T 1 triads are more connected to each other compared to other types. Furthermore, triads with low energy values have more tendency to be connected compared to high energy triads. While this pattern holds for both essential and nonessential gene networks, the essential network has more triads with the shared link. Besides modified explanations about the meaning of the energy-energy mixing pattern in the last paragraph of the “structural balance theory” subsection of the method section on page 7, in the result section on pages 10 and 11, the following steps that is taken to plot Fig 2 are added for further clarification. Moreover, to clarify Fig 5A more clearly, we added Fig 5B to it so that it can be easier for the readers to comprehend how each of the squares has been made. For each square:

1. The spectrums of energy of two specific types of triads are sorted.

2. Through moving on the energy axes, the number of triads that have a common link is counted and saved in a matrix in the Log scale.

3. The previous steps are repeated for all pair types of triads.

4. All 16 squares in 4 rows and 4 columns are merged.

Figure 2. An overview of creating connection between triads for one square (T 3 [+ + +] and T 2 [+ - +] triads).

About the signs of edges, as mentioned in the “data” subsection of the method section, the signed and weighted matrices studied here are presented publicly by Costanzo and his colleagues 2 . To better clarify how these signs are computed, a whole new paragraph to the “data” subsection of the method section, as the last paragraph on pages 3 and 4, is added describing the method by which signs are assigned to links by Costanzo et al 3 . Briefly speaking, when two genes are mutated, in terms of the size of the colony including them, the genetic interaction score (epsilon) between them is obtained. Thus, each gene has an interaction profile with other genes. By calculating the Pearson Correlation Coefficient (PCC) of these interaction profiles,

the genetic interaction similarity matrices have been provided.

2. Operationalization: It would be useful to explain further what constitutes an essential vs. nonessential gene; would be helpful to readers from other disciplines.

Response: We would like to thank the reviewer for the careful comment. We are committed to the definitions of essential and nonessential genes that Costanzo and et al. have proposed in their impactful study 3 . The features that specify whether a gene is essential or nonessential are the type of mutation, density (sparsity) of its corresponding network, the strength of interactions, the power of prediction in gene function, the biological process annotation. Thus, further explanation of what constitutes an essential vs. nonessential gene is added as an enrichment to the first paragraph of the “data” subsection of the method section, on page 3. 

3. Operationalization: It is still unclear how you define the "high impact" of significant genes (page 8)? Do you consider genes that sit on many ’balanced’ walks, or are there any measures you used to operationalize ’impact’?

Response: We appreciate the reviewer for the helpful comment. As Estrada 4 has proposed, according to Eq (3), the de- gree of balance of each given node (K i ) is calculated. In our study, we used the characteristic K i to find genes with the maximum value of K i in the network. These genes participate in the walks with only an even number of negative links; that is, those genes only participate in balanced walks. Since the value of K i is between zero and one; thus, we report the genes with K i = 1. As the term high impact may imply a sense of misunderstanding, we modified it in the manuscript on page 3, line 68. Besides, within the last paragraph of the method section, lines 284-287, on pages 7 and 8, we explained how the characteristic K i defines the degree of balance for a given node. Also, on page 8, as the respected reviewer has mentioned, we modified lines 274-275 (in the revised manuscript on page 11, lines 407-408).

Briefly, we changed the "high impact" term to "highest degree of balance" for each given gene with K i =1.

K i = exp A(Σ) / exp A(|Σ|) (3)

Last but not least, to contain more informative information, we edited the supplementary file and reported the genes with the highest degree of balance classified in terms of biological processes annotating.

4. Findings: You reported mean degree, the ratio between mean of squared degrees and squared of mean degree, modularity, assortativity coefficient, average path length, and clustering coefficient as indicators of a network’s topology. They are useful but not the only indicators of topology; explanations to why these particular measurements are used would be useful.

Response: Thanks to the reviewer for this insightful comment. As the respected reviewer has mentioned correctly, besides the reported features there are various indicators of network topology such as centrality, deformation ratio, robustness, etc, and we do have the same concern with the reviewer in this regard. Thus, among all network indicators calculated, we chose those reported in the manuscript based on the following three reasons. First, they are in line with our research question and the results we discussed. For example, modularity provides information on the networks’ communities, which is a very crucial feature in gene network studies 3 . As such, through computing the clustering coefficient we observe networks’ tendency to form triads, which are the basic building blocks in researches based on the balance theory. As well, the assortativity (disassortativity) indicator denotes the tendency of configurations to be connected with similar (different) counterparts. Second, some of the indicators have conceptual overlap with the studied features. For example, by calculating the mean degree we can have a sense of centrality. Third, other indicators such as robustness are useful for other research questions yet not directly related to our study. However, as the respected reviewer has mentioned, we calculated and added another useful feature to the revised manuscript, as below:

Small world propensity ( φ ): To quantify the extent to which a network displays a small-world structure, the Small-World Propensity, φ , is defined as below. ∆ C and ∆ L in Eq (4) show the deviation of clustering coefficient and path length, that are calculated as ∆ C = C latt −C / C latt −Crand and ∆ L = L − Lrand / L latt −Lrand , respectively, 

φ = 1 −√ ∆ C 2 + ∆ 2 / 2 (4)

Moreover, the following two indicators has been calculated as well, but not reported: Number of hubs (NHUBS): Nodes with degrees that exceed the average degree of the network are considered as hubs. Through this measurement according to Eq (5), not much more information would be presented in line with our aim. Thus, it is not reported in our study. Also, we were looking for genes that have a high degree of balance in the structure, not just those with high degrees. 

NHUBS = ∑ [k i > ⟨k⟩]. (5)

Synchronizability (S): This feature expresses the network’s power to synchronize and is calculated from the eigenvalues of the graph’s Laplacian matrix (Λ = D − A). In Eq (6), D defines the diagonal matrix containing the nodal degrees. As well, A is the unsigned adjacency matrix. The (S) is defined as the ratio between the first non-zero eigenvalue λ 2 and the largest eigenvalue λ max of Λ. While this indicator seems to prepare nice information, its insight as spectral analysis is far from our approach which is beyond pairwise interactions.

S =λ 2/ λ max (6)

The results of these features are presented in Table 1. 

Two other network’s features,

Number of hubs:

Essential 508(48.8462%) 

Nonessential 2131(48.1038%)

Synchronizability:

Essential 0.3127

Nonessential 0.1759

Thus, at the beginning of the “network analysis” subsection of the method section (page 4), we highlighted these informative, well-known, and standard indicators as has been similarly done by Barabási in his highly cited and seminal work. Besides, we added a new paragraph at the beginning of the result section on page 8 to explain the reason of selecting these indicators. Moreover, Fig 2 and Table 2 are modified regarding adding the small-worldness propensity ( φ ) feature.

5. Findings: In table 4, there’s an index value of 0 for genes-K shuffled cell. Can you explain the meaning of an index of 0 here?

Response: Thanks to the reviewer. Since the value of K Shu f f led is too small (i.e., 2.737e − 12 with mean = 2.952e − 12 and std = 5.631e − 13), it is displayed as 0.000. Specifically, according to Eq (7), K is a measure of the main difference that arises in the quantification of how close to balance an unbalanced network is,

K = trace exp A(Σ) / trace exp A(|Σ|) (7)

Moreover, for walks with all lengths, when we open the expansion of the signed matrix exp A(Σ) in Eq (7), among walks with all lengths, there can be some negative terms in the nominator, which represent an unbalanced walk. Though, in the denominator, all terms of expansion of the unsigned matrix exp A(|Σ|) are positive. Therefore, the walk balance index (K) flows between zero and one. Otherwise, if all present walks are positive, then this index meets its maximum value, which is one. The value closer to one implies more balance in the network, while close to zero is related to a largely unbalanced network. For example, when we shuffle the network, all structure is missing. Thus, it is inferable that when we have a random network, the walk balance index calculating the amount of balance in the network has the lowest possible value, that is, zero. For further clarification of this point in the manuscript, we explained this notion more precisely within the last paragraph of the method section, on page 7, lines 275-282, and the last paragraph of the result section on page 11 lines 398-401. Besides that K is nearly zero for the shuffled of the essential network, its value for the shuffled version of the nonessential network is 0.131. This minor difference here is due to the different percentages of positive (negative) links in both networks, which are also preserved in their corresponding shuffled networks. To be specific, the percentage of positive and negative links in the essential gene network are nearly equal ( 50.1% and 49.9% for positive and negative links, respectively). Thus, through shuffling this network, the equality between the signs results in a walk balance index too close to zero. Even though, in the nonessential gene network, the percentage of positive and negative links are different from each other. That is, there are 63.5% positive links, whereas the percentage of negative links is 36.4%.

6. Findings: Given the unequal sizes of the genes vs. nonessential genes network, with the nonessential genes network being much bigger, that could explain the higher ’likelihood’ for finding balanced configurations for this particular network. Do you consider controlling for the size of the networks, and/or normalization techniques to mitigate the sampling/size difference here?

Response: We thank the reviewer greatly for the comment. As the respected reviewer has mentioned, the size of the nonessential gene network is bigger than the essential one. But the fact is that there is more sparsity within the nonessential gene network compared to the essential network (Table 2). That is, the fraction of the number of actual links over the number of all possible ones in the nonessential gene network is 0.162, whereas in the essential gene network it is 0.461. Moreover, the density of triads in the nonessential gene network is less than the essential gene network. Specifically, the fraction of the number of triads over the number of all possible ones in the nonessential gene network is 0.006, yet in the essential gene network, it is 0.109. Therefore, it could explain the lower "likelihood" for finding balanced configurations for the nonessential gene network. In analyzing the balanced and unbalanced configurations for essential and nonessential networks according to the framework of structural balance theory, we compare each only with its corresponding shuffled network. As a result, we observe that balanced and unbalanced triads are over and underrepresented in both networks compared with the corresponding shuffled networks with the exact similar size for each. As such, Leskovec, according to Fig 3 in his highly referenced study, has performed this analysis for three different size networks, each with their shuffled ones 6 . As can be seen in Fig 3 (Table 1), there are three networks with different sizes, namely, Opinions, Slashdot, and Wikipedia. Then, in Table 2, p(T i ) and p 0 (T i ) are introduced as the probability of each type of triad in original and shuffled networks, respectively. Finally, in Table 3, the result of these quantities, which are independent of the networks’ sizes, are reported. In other words, these networks are compared with their shuffled corresponding ones, which have the same sizes, not each other.

Figure 3. Comparison of three different size networks, each with their shuffled ones in terms of all four types of triads 6 .

About our analysis on walks with all lengths, it is worth mentioning that the (K) as the density of balanced walks with all lengths with values between zero and one, is independent of size. Therefore, the greater this index for a network is, the more balanced it is. Besides that, similar to what has been concluded while considering triads, through analyzing the walk balance index (K), both networks are more balanced than their corresponding shuffled ones. Estrada has performed the same comparison in his study 4 . Therefore, we have completed all three sentences in the manuscript where being more balanced of nonessential than the essential network is told. We applied this important point within the last sentence of the abstract section, on page 1, the result section, on page 11, lines 401-405, and the discussion section, on pages 12 and 13, lines 446-449. At last, especially thanks to the respected reviewer, according to normalized indicators which are dependent on size (clustering coefficient and average path length) in the revised manuscript, we have added three new paragraphs, including three new equations and references to the “network analysis” subsection of the method section, on page 4 and 5. Furthermore, the second paragraph of the result section on page 8, Fig 2, and Table 2 are modified as well.

References

1. Moradimanesh Z, Khosrowabadi R, Eshaghi Gordji M, Jafari GR. Altered structural balance of resting state networks in autism. Scientific Reports. 2021; 11:1966. https://doi.org/10.1038/s41598-020-80330-0 [Ref 43 in the manuscript]

2. http://boonelab.ccbr.utoronto.ca/supplement/costanzo2016/

3. Costanzo M, et al. A global genetic interaction network maps a wiring diagram of cellular function. Science. 2016; 353:6306. https://doi.org/10.1126/science.aaf1420 PMCID: PMC5661885 [Ref 48 in the manuscript]

4. Estrada E, Benzi M. Walk-based measure of balance in signed networks: Detecting lack of balance in social networks. Physical Review E. 2014; 90:042802. https://doi.org/10.1103/PhysRevE.90.042802 PMCID: 25375544 [Ref 47 in the manuscript]

5. Barabasi AL, Oltavi ZN. Network biology: understanding the cell’s functional organization. Nature Reviews Genetics. 2004; 5(2):101-113. https://doi.org/10.1038/nrg1272 PMID: 14735121 [Ref 1 in the manuscript]

6. Leskovec J, Huttenlocher D, Kleinberg J. Signed networks in social media. Proc. SIGCHI Conf. Hum. Factors Comput. Syst. 2010; 22:13611370. https://doi.org/10.1145/1753326.1753532 [Ref 3 in the manuscript]

---

## [Decision Letter · Decision Letter 1]

10 Mar 2022

The structure balance of gene-gene networks beyond pairwise interactions

PONE-D-21-31371R1

Dear Dr. Jafari,

We’re pleased to inform you that your manuscript has been judged scientifically suitable for publication and will be formally accepted for publication once it meets all outstanding technical requirements.

Note that one reviewer has a minor suggestion with regard to weak balance theory that you may choose to incorporate.

Kind regards,

Bryan C Daniels

Academic Editor

PLOS ONE

Additional Editor Comments (optional):

Reviewers' comments:

Reviewer's Responses to Questions

**Comments to the Author**

1. If the authors have adequately addressed your comments raised in a previous round of review and you feel that this manuscript is now acceptable for publication, you may indicate that here to bypass the “Comments to the Author” section, enter your conflict of interest statement in the “Confidential to Editor” section, and submit your "Accept" recommendation.

Reviewer #1: All comments have been addressed

Reviewer #2: All comments have been addressed

2. Is the manuscript technically sound, and do the data support the conclusions?

Reviewer #1: Yes

Reviewer #2: Yes

3. Has the statistical analysis been performed appropriately and rigorously? 

Reviewer #1: Yes

Reviewer #2: Yes

4. Have the authors made all data underlying the findings in their manuscript fully available?

Reviewer #1: Yes

Reviewer #2: Yes

5. Is the manuscript presented in an intelligible fashion and written in standard English?

Reviewer #1: Yes

Reviewer #2: Yes

6. Review Comments to the Author

Reviewer #1: (No Response)

Reviewer #2: Thank you for addressing my comments and making the appropriate changes to the manuscript.

One minor suggestion in terms of conceptualization of strongly vs. weakly balanced triads: while you have cited Davis' work which is foundational to the understanding SBT, I believe the triad T0 (- - -) should actually be a weakly balanced triad (under Davis' Weak Balance Theory), as opposed to weakly unbalanced. Thus, with this definition, triads T0, T1, and T3 are all balanced, and triad T2 (+ + - ) is the only unbalanced triad type. I would recommend you to revisit the tenets of Weak Structural Balance Theory (WSBT) if you would still want to include this version of structural balance theory in your conceptualization.

A way to ensure that you're consistent with different version of structural balance theory is to clarify what properties of balance you are considering for this specific network context, and your own rationale for conceptualizing "strong" vs. "weak" balance in this manner.

7. PLOS authors have the option to publish the peer review history of their article (what does this mean?). If published, this will include your full peer review and any attached files.

Reviewer #1: No

Reviewer #2: No

---

## [Editor Report · Acceptance letter]

21 Mar 2022

PONE-D-21-31371R1 

The structure balance of gene-gene networks beyond pairwise interactions 

Dear Dr. Jafari:

I'm pleased to inform you that your manuscript has been deemed suitable for publication in PLOS ONE. Congratulations! Your manuscript is now with our production department. 

Kind regards, 

on behalf of

Dr. Bryan C Daniels 

Academic Editor

PLOS ONE